# What are the experiences of teleophthalmology in optometric referral pathways? A qualitative interview study with patients and clinicians

Dilisha Patel ,[1] Sarah Abdi ,[1] Josie Carmichael,[1,2] Konstantinos Balaskas,[2] Ann Blandford [1]

[1]UCL Interaction Centre, University College London, London, UK
[2]Moorfields Eye Hospital NHS Foundation Trust, London, UK

**Correspondence to**
Professor Ann Blandford;
a.blandford@ucl.ac.uk

## ABSTRACT

**Objective** Implementing teleophthalmology into the optometric referral pathway may ease the current pressures on hospital eye services caused by over-referrals from some optometrists. This study aimed to understand the practical implications of implementing teleophthalmology by analysing lived experiences and perceptions of teleophthalmology in the optometric referral pathway for suspected retinal conditions.

**Design** Qualitative in-depth interview study

**Setting** Fourteen primary care optometry practices and four secondary care hospital eye services from four NHS Foundation Trusts across the UK.

**Participants** We interviewed 41 participants: patients (17), optometrists (18), and ophthalmologists (6) who were involved in the HERMES study. Through thematic analysis, we collated and present their experiences of implementing teleophthalmology.

**Results** All participants interviewed were positive towards teleophthalmology as it could enable efficiencies in the referral pathway and improve feedback and communication between patients and healthcare professionals. Concerns included setup costs for optometrists and anxieties from patients about not seeing an ophthalmologist face to face. However, reducing unnecessary visits and increasing the availability of resources and capacity were seen as significant benefits.

**Conclusions** Overall, we report positive experiences of implementing teleophthalmology into the optometric referral pathway for suspected retinal conditions. Successful implementation will require appropriate investment to set up and integrate new technology and remunerate services, and continued evaluation to ensure timely feedback to patients and between healthcare professionals is received.

**Trial registration number** ISRCTN18106677.

## STRENGTHS AND LIMITATIONS OF THIS STUDY

⇒ The strengths of our study include our ability to report on participants' rich and compelling lived experiences through in-depth qualitative interviews of using and being involved in referral pathways that used teleophthalmology over prospective opinions about the technology.

⇒ We can further qualify previous literature that has reported teleophthalmology's potential impacts and benefits with these lived experiences within the HERMES study.

⇒ This study collected data from multiple stakeholders, including primary and secondary eye care professionals and patients.

⇒ A limitation of this study was that all ophthalmologists interviewed were involved in the HERMES study and, therefore may have had more knowledge of teleophthalmology than others.

## INTRODUCTION

Primary eyecare in the UK is mainly delivered by community optometry practices.[1 2] If patients are suspected of having a retinal condition, referrals to hospital eye services (HES) are typically processed by their general practitioner (GP) based on recommendations from the community-based optometrist. Thus, optometrists are often not involved in making direct referrals using electronic referral platforms or informed of outcomes. The additional step can reduce the specificity of clinical details included in the referral, as GPs are not specialists in eye care and rarely have the time or expertise to undertake eye examinations.[3] For several reasons, including concerns over capacity, changes in practice due to the COVID-19 pandemic, and the high number of referrals to HES, there is a need for disruptive changes in the optometric referral pathways for suspected retinal conditions (SRC).[4] We explore the experiences of patients, optometrists, and ophthalmologists of teleophthalmology for SRC.

Teleophthalmology describes the process of providing health information with medical

technology at a distance, geographical, temporal or both,[5 6] to facilitate decision-making. In our study, we focused on an asynchronous platform; this allowed later review, through the uploading of clinical information and multimodal retinal imaging, that is, fundus photography and macular optical coherence tomography (OCT) scans directly from the optometry practice, by the receiving HES-based ophthalmologists. A benefit of the custom-built study platform, when compared with recent attempts for the commissioning of teleophthalmology-like store-and-forward referral pathways, is that it enables optometrists to upload full-volume OCT scans from multiple device manufacturers, in both their proprietary and open-source file format, which can then be parsed and reconstructed into a high-quality full volume scan, directly viewable on the platform's embedded viewer.[7 8] Given the critical role of OCT imaging for diagnosing and managing medical retina conditions, a minimum clinical data set including full-volume OCT is a prerequisite for the safe and efficient delivery of teleophthalmology referral triaging, providing ophthalmologists with essential information to confidently make referral triaging decisions. In the absence of teleophthalmology, or when attempts to implement teleophthalmology-like pathways are made with minimal input from clinical informatics and HES-based clinician and imaging experts, triaging ophthalmologists receive referrals, either without any accompanying imaging or frequently with one or a few selected cross-sectional images, not the entire volume,[9] then referred patients are seen face to face to assess whether further investigation or treatment is needed.

The non-urgent referral pathway with and without teleophthalmology is presented in figure 1. A recent systematic review reported that teleophthalmology and digital referrals could reduce waiting time, costs, and unnecessary referrals. It also noted that teleophthalmology could lead to earlier detection and diagnosis[10] and as such is an underutilised resource for HES. This review was based on reviewing referrals, not people's first-hand experiences. Therefore, there is a need to understand how implementation would affect users in practice.

Reports have shown that the growing use of OCT scanning has increased the number of referrals to HES;[11 12] therefore, teleophthalmology has been suggested to reduce unnecessary referrals, manage growing capacity concerns, and potentially manage increasing workloads by reviewing referrals before patients are seen in clinics.

We present findings from a study linked with a cluster randomised clinical trial (HERMES) evaluating the effectiveness of a teleophthalmology platform.[1] Those in the intervention arm of the trial were using teleophthalmology to refer patients to HES, while those in the control arm were using their regular referral pathways. We report the novel qualitative findings of patients' and healthcare professionals' experiences to understand the practical implications of implementing teleophthalmology.

## MATERIALS AND METHOD

This study's overall aims, objectives, and recruitment strategy are detailed elsewhere;[13] we summarise key points here. We undertook semistructured interviews with 41 participants from across the UK, between December 2021 and May 2022. All participants were recruited from sites participating in the wider HERMES trial. These comprised six ophthalmologists who were all making remote referral decisions; 18 community optometrists, of whom 12 were from the intervention arm and six from the control arm of the HERMES study; and 17 patients recruited from community practices, of which 14 were from clinics in the intervention arm and three from clinics in the control arm. These sites were affiliated with four NHS Trusts; Moorfields Eye Hospital NHS Foundation Trust, Central Middlesex Hospital at London North West University Healthcare NHS Trust, Birmingham University Hospital NHS Foundation Trust and North West Anglia NHS Foundation Trust. A semistructured topic guide (attached in online supplemental material) was used for all interviews and was tailored to each participant group. All interviews were audiorecorded with consent. Five interviews were manually transcribed verbatim to aid familiarisation; the remaining interviews were transcribed using Scrintal Software. All transcripts were anonymised and checked for

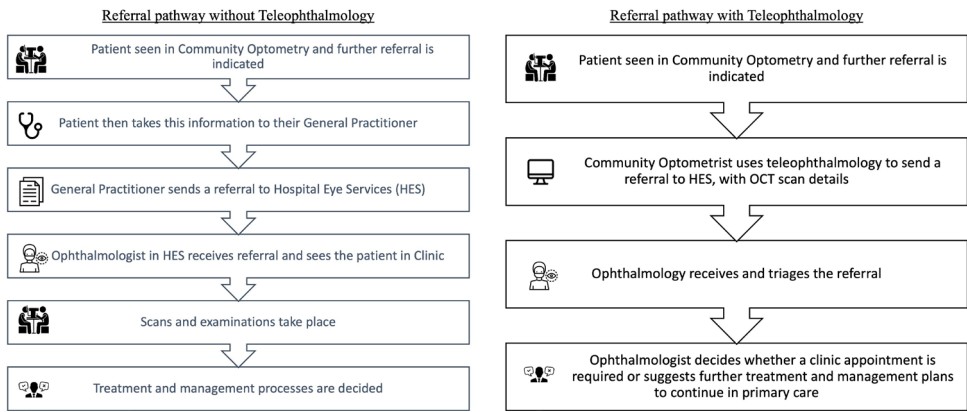

**Figure 1** The non-urgent patient referral pathway with and without teleophthalmology. OCT, optical coherence tomography.

accuracy. The transcripts were coded using NVivo by an independent researcher who did not conduct the interviews, using inductive thematic analysis methods.[14 15] Of note, 35 codes were initially defined and discussed with the research team. These were then refined and categorised into overarching themes. We present the three main themes which explore experiences of teleophthalmology for referrals for SRC.

### Patients and public involvement

Eighteen patients were consulted in the planning and development of the wider HERMES study, which has been detailed in the protocol paper.[13] The insights raised about understanding the benefits of teleophthalmology contributed to the design of this qualitative study. As noted above, 17 patients participated in interviews.

## RESULTS

We present our findings under three themes, *Efficiencies of Teleophthalmology; Teleophthalmology enables Feedback;* and *Concerns about Teleophthalmology.* We found most participants were optimistic about the implementation of teleophthalmology in the optometric referral pathway due to the efficiencies the platform would enable. All participants expressed needing feedback during the referral process to improve care and highlighted some concerns.

### Efficiencies of teleophthalmology

All welcomed teleophthalmology due to its ability to improve patient and clinician experiences. There is regional variation in referral pathways depending on the specific condition; GPs typically process routine referrals for SRC. However, it was reported that GPs were not always suited to create referrals due to their limited skillset in specialist eyecare. Patients often described GPs as the unnecessary 'middleman'. Patients reported being keen to be referred directly by their optometrist and felt this would reduce their waiting time to hear back from HES if referrals were processed directly.

> If the optician can do the referral directly rather than you know, the optician telling me you'll need to go and see your GP who will refer you (…) I would be more comfortable because they (optometrist) know what they're doing whereas the GP is just saying you are alright then, if your optician told you that, then I'll send you. (Patient 11)

In the HERMES study, optometrists used teleophthalmology to refer patients to HES, enabling a quicker referral process direct to triaging ophthalmologists. They shared that teleophthalmology could improve patient satisfaction and help relieve hospital capacity pressures by reducing unnecessary hospital visits.

> I think the whole teleophthalmology thing will improve patient satisfaction and it makes life a lot easier, less patients, elderly patients having to find transport to the hospital, and being dilated once in the optometry practice and then being dilated again, back at the hospital and patient transport having to be arranged, so overall good, big saving of cost and finance and less crowded waiting rooms at the hospital. (Optometrist 18)

Ophthalmologists shared that the teleophthalmology platform introduced uniformity to the referral process by requiring the same data fields to be completed for each patient, which was easy for the referring optometrist. This enabled referrals to be triaged and reviewed more quickly: referral decisions could be made promptly, and the appropriate triaging decision could be made regarding the indication and the urgency for a hospital visit.

> It's more informative because, as you know, the platform has the questions with the tick box, um, on the optician findings which (is) not always involved in a classic referral proforma. And obviously, it has the imaging as well, which helps us to make a decision very quickly. (Ophthalmologist 1)

As mentioned, the key benefit of the teleophthalmology platform is the ability to review and triage patient referrals without them having to attend a face-to-face appointment. All participants recognised and shared the advantage of saving time and resources by using teleophthalmology.

### Teleophthalmology enables feedback

Some patients reported that they were happy for teleophthalmology to be used to assess their referrals as they would not want to attend HES if not required. Still, they expected feedback to explain the reason for not being seen for a face-to-face appointment. This was not always provided. Some patients reported dissatisfaction with their referral experience due to the lack of communication. This was also shared in the context of not receiving feedback promptly. Patients expected to hear about their referral decision more promptly through the teleophthalmology process, which increased concerns over their eye health when this expectation was not satisfied. In these cases, patients wanted to be seen or told directly and promptly why an appointment was not required and not to be kept waiting in uncertainty. Teleophthalmology can overcome this expectation discrepancy through accurate information presentation and timely and clear feedback.

> If the patients were to get a letter or some form of communication from the hospital or the specialist to reassure them that your case has been looked at and this is what has been concluded, I think that would be enough to put somebody's mind at ease. (Patient 10)

Optometrists stated that one of the significant benefits of teleophthalmology was the ability to receive feedback from ophthalmologists. Optometrists reported that when patients were referred to HES in the absence of teleophthalmology, patients would often return to them seeking more information and advice about their care; therefore,

it was important for optometrists to be involved in the referral pathway and remain informed of their patients' management plans. Optometrists shared that many patients were not a reliable source of information about their eye health/treatment, which could affect future care or monitoring they provided.

> Once we refer the patient, we don't actually know then what is happening thereafter, unless we chase the patient or, our patients are quite loyal, so they would, we would see them a year later, we will say or remember, we referred you last time, what happened? (…) So, it's actually we're basing it off what the patient is then telling us, so we are actually getting like the second story through the patient rather than the actual clinical information. (Optometrist 12)

By receiving feedback, optometrists can also verify whether their referrals were appropriate and audit themselves to improve the quality of their referrals.

> Because if we keep referring something that we think is urgent, but (the) ophthalmologist tells us this is not urgent, and if you learn by that, that's going to help you, you see. Right now, there's no feedback (…). But if I got feedback from the ophthalmologist that saw the patient and I will know for next time when I see that similar sort of situation that well, actually this isn't urgent. (Optometrist 10)

Having a system where the community-based clinic is connected to the HES was also seen as a great benefit for ophthalmologists. They welcomed being able to provide feedback to the referring optometrists, especially to enable the sharing of referral decisions directly and concurred with the need to provide feedback on the referral quality to improve future referrals. It was suggested that the teleophthalmology system should send referral replies to the referring optometrist, patient, and GP so all are informed of the outcome.

### Concerns about teleophthalmology

There were some concerns about using technology to manage patient referrals. Some patients still wanted the reassurance of seeing a clinician rather than having their referral decision and notification completed remotely. Seeing someone face to face provided the holistic care some patients reported wanting and addressed their worries and anxieties.

> I think (if) you don't get a chance to see the patient yourself, there is something about looking at data visually transactionally, that is fine, but there is also something about talking to the patients about how they're feeling and how they're coping with things. (Patient 10)

Optometrists were mainly concerned with the practicalities of implementing a new system into their workflow. This included concerns over training to use the equipment, the reliability of network connectivity, and equipment costs that some smaller practices may not be able to bear, as well as remuneration for their time for taking on additional roles. Some also reported that completing a referral on the teleophthalmology platform took time.

> The barriers would be cost, because this is all based on the information that is being sent from an OCT device, yeah, as part of the process of referral, it's not just from a letter, so when it comes to having the equipment, that's an immediate barrier. And having the right remuneration for the equipment. (Optometrist 2)

Ophthalmologists also shared these concerns; however, they were positive towards the ability of teleophthalmology, enabling them to use their time more efficiently.

## DISCUSSION

While previous work has focused on the efficacy and efficiency of teleophthalmology platforms through reviewing referrals,[12 16] we report insights based on experiences from patients, optometrists, and ophthalmologists to validate previous findings on perceptions of using teleophthalmology for SRC.

All participants recognised the value of implementing a teleophthalmology system into their ophthalmic care pathway due to its potential to improve patient care and health services efficiencies. This is supported by others,[17] who found through a review of referrals that teleophthalmology can reduce unnecessary HES visits and significantly impact patient anxieties.[18] Our study has shown this in practice, with many patients sharing that they would not want to attend HES if not required.

Both optometrists and ophthalmologists reported that teleophthalmology's significant advantage is the ability to electronically refer patients directly from optometrist practices to HES, which can significantly reduce the waiting time for patients. The ability to triage referrals electronically also enabled ophthalmologists to provide replies and feedback to the referring optometrist via the teleophthalmology platform, which they greatly valued.

Implementing teleophthalmology into the eye care pathway would remove the burden on GPs of having to process patient referrals, but they must be informed of such referrals. GPs have, in principle, supported the suggestion of optometrists referring patients directly,[16 19] and we found that patients and optometrists would support this change in practice. There is great value in involving optometrists in the referral process, as it has been reported that this could reduce unnecessary referrals by approximately half[12]

The overarching theme shared by all participants which substantiates many of the benefits of the teleophthalmology platform, is the potential of the platform to facilitate the provision of feedback. The importance of receiving timely feedback in eye care, in general, has been reported by others[3 20] and was essential for patients

to alleviate their concerns over their eye health. Harvey *et al*[20] specifically outline the factors that could affect the provision of feedback, and a key implication of their work is the call for technology to support this provision. Thus, we found that teleophthalmology could overcome concerns in the optometric referral pathway.

While feedback can keep both patients and optometrists informed,[21] it can also improve future referral quality through open conversations between referring clinicians. Our results concur as we report optometrists greatly value receiving referral replies directly from ophthalmologists to remain informed of their patients' care and audit their referrals. Additionally, previous research has highlighted that the lack of communication between optometrists and ophthalmologists can be problematic;[22 23] therefore, implementing a teleophthalmology platform could help to overcome this.

Potential barriers raised by some optometrists were the initial setup costs, which include time, training, and financial costs. While others have reported this should be considered,[10 16] we found this to be a key concern in practice. According to the current General Ophthalmic Services contract (2023), OCT scans are not a contracted service in optometric care.[24] Therefore, there is a need to ensure that optometrists are appropriately remunerated for providing this service. Optometrists would also need to have appropriate access to NHS e-referral systems to expediently refer patients to the HES system. Others have begun to explore the cost-effectiveness of implementing teleophthalmology systems;[25] further work is needed to establish health-economic benefits concerning SRC that were raised in our study.

To successfully implement teleophthalmology into the optometric referral pathway, there needs to be an investment to enable parity among optometry practices to support new technology setup. While the implementation of teleophthalmology was perceived to initially increase the workload of optometrists to process referrals and ophthalmologists to triage referrals, with the correct remuneration, there is significant potential to relieve pressures on stretched eye care services.

We acknowledge study limitations including the influence of participation bias. The participants who chose to participate in our research may have different views from those who did not participate, which have not been captured in our study. We also recognise that the ophthalmologists who chose to participate were involved in the HERMES study itself, which led to increased knowledge of teleophthalmology. Future work should endeavour to recruit a more diverse sample of participants to capture broader views on the experiences of teleophthalmology. However, the strengths of our study include the use of in-depth interviews, through which we were able to elicit the lived experiences of those involved in the study, many of whom had direct experience with the teleophthalmology platform. We were also able to recruit participants across the stakeholder group, thus providing multiple perspectives on the teleophthalmology pathway.

Optometrists and ophthalmologists provided their perspectives on using the platform and the practicalities involved in use, while patients reported on the personal impacts of navigating their eye-health journey through teleophthalmology. These diverse perspectives have enabled us to corroborate and extend existing understanding of the practical implications of implementing teleophthalmology.

## CONCLUSIONS

Implementing teleophthalmology into the optometric referral pathway has numerous benefits, as outlined by all participants. Through our in-depth interview study with patients, optometrists, and ophthalmologists, we found that they generally report great value in implementing teleophthalmology through improving efficiency and the ability to provide and receive feedback. Patients were satisfied if their referrals were reviewed with teleophthalmology to reduce the possibility of unnecessary HES visits if this was clearly and efficiently communicated back to them. Optometrists felt they were better suited than GPs to write and process patient referrals and would feel more valued if they were more directly involved in the pathway. Finally, ophthalmologists were pleased with a system enabling them to manage their caseloads more efficiently. Further efficiencies teleophthalmology can promote include removing the burden on GPs, the time patients wait to be seen by HES, the time it takes for ophthalmologists to review and provide referral replies and finally, the overarching benefit to all participants of being involved and receiving feedback. Future work could explore how to overcome barriers such as connectivity and the specific health economics of implementing teleophthalmology to validate our findings.

**Acknowledgements** We thank all those involved in this research, including participants and the wider research team.

**Contributors** DP analysed the data and prepared the manuscript. SA, KB, and AB designed the study protocol, and SA undertook recruitment and data collection. SA, JC, and AB contributed analysis as presented by DP. All authors reviewed and contributed to subsequent drafts and approved the final manuscript. AB is the overall guarantor for this work.

**Funding** This work was funded by NIHR Health Technology Assessment grant number 18/182.

**Competing interests** None declared.

**Patient and public involvement** Patients and/or the public were involved in the design, or conduct, or reporting, or dissemination plans of this research. Refer to the Methods section for further details.

**Patient consent for publication** Not required.

**Ethics approval** This study involves human participants and was approved by Health Research Authority. London-Bromley Research Ethics Committee (20/LO/1299). Participants gave informed consent to participate in the study before taking part.

**Provenance and peer review** Not commissioned; externally peer reviewed.

**Data availability statement** Data are available upon reasonable request. We are happy to be contacted to share anonymised data. Please contact the corresponding author for requests.

**ORCID iDs**
Dilisha Patel http://orcid.org/0000-0002-3746-8171
Sarah Abdi http://orcid.org/0000-0002-4395-8257
Ann Blandford http://orcid.org/0000-0002-3198-7122

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
