## [Reviewer comments · BMJ Open]

ARTICLE DETAILS

TITLE (PROVISIONAL)	What are the Experiences of Teleophthalmology in Optometric Referral Pathways?: A Qualitative Interview Study with Patients and Clinicians.
AUTHORS	Patel , Dilisha; Abdi, Sarah; Carmichael, Josie; Balaskas, Konstantinos; Blandford, Ann

VERSION 1 – REVIEW

REVIEWER	Parkins, David Bexley Care Trust
REVIEW RETURNED	17-Oct-2023

GENERAL COMMENTS	Thank you for the opportunity to review this paper. It is timely with the national focus on improving communications between community optometrists and hospital eye services (HES), and reducing unnecessary HES visits. I offer the following observations consideration. It is worth noting the context in which OCT is offered to most patients in primary optical/optometric practices. It is a private transaction and not mandated under General Ophthalmic Services (GOS). The NHS sight test is a single episode of care and does not fund the optometrist to review or monitor patients, or fund when seeking advice and guidance for possible continuing management. Referral requirements under the GOS contract - https://www.england.nhs.uk/wp-content/uploads/2018/08/PRN00242-General-ophthalmic-mandatory-services-model-contract-September-2023.pdf involves: 31. Where the Contractor or an ophthalmic practitioner employed or engaged by it to perform the Contract is of the opinion that a patient whose sight has been tested pursuant to clause 30— 31.1. shows on examination signs of injury, disease or abnormality in the eye or elsewhere which may required medical treatment; or 31.2. is not likely to attain a satisfactory standard of vision notwithstanding the application of corrective lenses, it shall, if appropriate, and with the consent of the patient— 31.3. refer the patient to an ophthalmic hospital, which includes an ophthalmic department of a hospital, 31.4. inform the patient’s doctor or GP practice that it has done so, and 31.5. give the patient a written statement that it has done so, with details of the referral Consequently, a teleophthalmology advice and guidance service using OCT file transfer may need to be funded. Since 2018, all routine and urgent referrals from GPs have to be submitted on
--

	NHS eRS. The vast majority of optical and optometric practices do not have access to NHS eRS. However, for ICB commissioned enhanced services, practices may have access to 3rd party referral software platforms which link to NHS eRS. Other barriers are lack of connectivity within primary optical / optometric practices and interoperability of referral platforms with practice systems (e.g., double entry required for referral process and for recording advice and guidance replies in clinical records) and hospital systems. General comments Page 2, line 11. What is already known about on this topic - Many patients waiting to be seen are unnecessary referrals. There is unwarranted variation in referrals by some optometrists which results in this typical view from the hospital perspective. As optometrists are not contracted under the General Ophthalmic Services (GOS) to refine, manage or monitor, they may have no alternative but to refer. Suggestion ... Some patients waiting to be seen may not require a face-to-face referral; ... Page 2, line 13. Consider suggestion ... feel 'undervalued by hospital eye services and uninformed about their patients care'. Abstract Page 2 line 27. Objective: It is more accurate to say ... by over-referrals from some optometrists. Evidence shows that there is unwarranted variation by a smaller number of outliers. Page 2, line 37. Results: comment - teleophthalmology may not by itself reduce 'unnecessary referrals', as referral decisions are multifactorial. Consider rewording However, 'reducing unnecessary HES visits and as this is reported in the text. It is possible that the ease of getting a second opinion may increase optometrist referrals with requests for teleophthalmology advice and guidance, and these can be managed more appropriately. Page 2, line 41. Suggest ... ensure 'timely' feedback to patients ... (links to page 5, lines 9 to 45). Page 2, line 54. Strengths and Limitations - Another limitation applies to the optometrists in that they have all chosen to participate in the clinical trial, and being an engaged group may not represent the views of the wider body of practices e.g., corporate optical/ optometric organisations. Introduction Page 3 line 5. 'Primary eyecare in the U.K. is mainly delivered by community optometry practices'. This is referenced to a sentence in Han et al. 2022, but there is no reference against this. Blandford et al.2022 quoted Evans et al. Referrals from community optometrists to the hospital eye service in England. Ophthalmic Physiol Opt 2021;41:365–77. for a similar statement. Evans et al. source reference is Bowling B, Chen SD, Salmon JF. Outcomes of referrals by community optometrists to a hospital glaucoma service. Br J Ophthalmol. 2005;89:1102–04. Page 3, line 7. ... actual referral pathway ... - For GOS only practices, routine referral usually involves the GP, but where primary eye care services are commissioned, optometrists already use direct electronic referral platforms, however, these do not always provide a route back for referral outcome letters. Suggest revised wording: 'Thus, some optometrists are not activity involved in making direct referrals using electronic referral platforms or informed of referral outcomes. Page 3, line 10. Comment – I agree the GP step can reduce specificity of clinical details, with reduced clinical information and
--	--

	even the optometrist referral letter omitted, and the process does not facilitate optometrist image transfer. Page 3, line 23. A lot has changed since C Roberts blog 2021, consider new reference page https://transform.england.nhs.uk/key-tools-and-info/digital-playbooks/eye-care-digital-playbook/improve-referrals-to-eye-care-services/ Page 3, line 32. The systematic review conclusions demonstrated that optometrist-facilitated teleophthalmology can dramatically reduce referrals and streamline care. In addition, the increasing prevalence of OCT in optometric practice represents an underutilized resource for HES. Page 4. No comments Page 5, line 6. Still, they expected feedback to explain the reason for non-referrals. The optometrist has made a referral, the HES has reviewed the OCT scan and decided that there is no need for the patient to be seen face to face. This is not a non-referral. Consider rewording: Still, they expected feedback to explain the reason for not being seen for a face-to-face visit. Page 5, line 9-45 Comment: some really important messages around timely feedback to patient and referral learning feedback loop to optometrists. Discussion Page 6, line 25. Suggest ... reduce unnecessary HES visits and Page 6, line 35. Please reference NHS planning guidance 2023/24, where the key actions are to expand direct access and self-referral where GP involvement is not clinically necessary. Integrated care system commissioners are asked to put in place direct referral pathways from community optometrists for all urgent and elective care consultations by September 2023. This is to release capacity in GP practices. NHS England. Priorities and operational-planning-guidance 2023/2024. https://www.england.nhs.uk/wp-content/uploads/2022/12/PRN00021-23-24-priorities-and-operational-planning-guidance-v1.1.pdf Page 7, line 25. Comment: further work should also explore connectivity and interoperability with primary and secondary care systems.
--	--

REVIEWER	Turner, Angus Lions Eye Institute
REVIEW RETURNED	28-Nov-2023

GENERAL COMMENTS	This is helpful qualitative research as part of a broader trial. I think the reader needs to be clear from the beginning about the narrow scope of the teleophthalmology in question; ie. name the process as asynchronous triage based on macular multi-modal imaging. Since there are many modes of teleophthalmology, it's helpful for the reader to be more informed of the system being investigated for the context of the patient and clinician experience to follow. Small point is long sentences; Pg 3 paragraph beginning line 15 and pg 6 paragraph beginning line 28 as 2 examples leading to loss of clear meaning being conveyed.
---

VERSION 1 – AUTHOR RESPONSE

Reviewer 1 comments	
It is worth noting the context in which OCT is offered to most patients in primary optical/optometric practices. It is a private transaction and not mandated under General Ophthalmic Services (GOS). The NHS sight test is a single episode of care and does not fund the optometrist to review or monitor patients, or fund when seeking advice and guidance for possible continuing management. Referral requirements under the GOS contract - https://www.england.nhs.uk/wp-content/uploads/2018/08/PRN00242-General-ophthalmic-mandatory-services-model-contract-September-2023.pdf involves: 31. Where the Contractor or an ophthalmic practitioner employed or engaged by it to perform the Contract is of the opinion that a patient whose sight has been tested pursuant to clause 30— 31.1. shows on examination signs of injury, disease or abnormality in the eye or elsewhere which may required medical treatment; or 31.2. is not likely to attain a satisfactory standard of vision notwithstanding the application of corrective lenses, it shall, if appropriate, and with the consent of the patient— 31.3. refer the patient to an ophthalmic hospital, which includes an ophthalmic department of a hospital, 31.4. inform the patient’s doctor or GP practice that it has done so, and 31.5. give the patient a written statement that it has done so, with details of the referral Consequently, a teleophthalmology advice and guidance service using OCT file transfer may need to be funded. Since 2018, all routine and urgent referrals from GPs have to be submitted on NHS eRS. The vast majority of optical and optometric practices do not have access to NHS eRS. However, for ICB commissioned enhanced services, practices may have access to 3rd party referral software platforms which link to NHS eRS. Other barriers are lack of connectivity within primary optical / optometric practices and interoperability of referral platforms with practice systems (e.g., double entry required for referral process and for recording advice and guidance replies in clinical records) and hospital systems.	Thank you for these valuable insights. We have added this reference to the discussion and implications of implementing teleophthalmology into practise, including the consideration of connectivity.

Page 2, line 11. What is already known about on this topic - Many patients waiting to be seen are unnecessary referrals. There is unwarranted variation in referrals by some optometrists which results in this typical view from the hospital perspective. As optometrists are not contracted under the General Ophthalmic Services (GOS) to refine, manage or monitor, they may have no alternative but to refer. Suggestion ... Some patients waiting to be seen may not require a face-to-face referral;	As per journal guidelines we have removed this section.
Page 2, line 13. Consider suggestion ... feel 'undervalued by hospital eye services and uninformed about their patients care'.	As per journal guidelines we have removed this section.
Page 2 line 27. Objective: It is more accurate to say ... by over-referrals from some optometrists. Evidence shows that there is unwarranted variation by a smaller number of outliers.	We have changed this as per your suggestion.
Page 2, line 37. Results: comment - teleophthalmology may not by itself reduce 'unnecessary referrals', as referral decisions are multifactorial. Consider rewording However, 'reducing unnecessary HES visits and as this is reported in the text. It is possible that the ease of getting a second opinion may increase optometrist referrals with requests for teleophthalmology advice and guidance, and these can be managed more appropriately.	Thank you we appreciate this feedback. We have recently completed the statistical analysis of the main HERMES trial, and teleophthalmology alone was shown to statistically significantly reduce the number of unnecessary referrals (alternatively phrased: increases the number of patients deemed not to require a hospital visit) by 15% and reduces the number of patients requiring urgent referrals (downgrading them to routine referrals) by 60%. We have reworded as suggested, and the above will be published soon.
Page 2, line 41. Suggest ... ensure 'timely' feedback to patients ... (links to page 5, lines 9 to 45).	Changed, as suggested.
Page 2, line 54. Strengths and Limitations - Another limitation applies to the optometrists in that they have all chosen to participate in the clinical trial, and being an engaged group may not represent the views of the wider body of practices e.g., corporate optical/ optometric organisations.	We have included participation bias in the limitations now.
Page 3 line 5. 'Primary eyecare in the U.K. is mainly delivered by community optometry practices'. This is referenced to a sentence in Han et al. 2022, but there is no reference against this. Blandford et al.2022 quoted Evans et al. Referrals from community optometrists to the hospital eye service in England. Ophthalmic Physiol Opt 2021;41:365–77. for a similar statement. Evans et al. source reference is Bowling B, Chen SD, Salmon JF. Outcomes of referrals by	Thank you for these references, we have included the source citation to this point.

community optometrists to a hospital glaucoma service. Br J Ophthalmol. 2005;89:1102–04	
Page 3, line 7. ... actual referral pathway ... - For GOS only practices, routine referral usually involves the GP, but where primary eye care services are commissioned, optometrists already use direct electronic referral platforms, however, these do not always provide a route back for referral outcome letters. Suggest revised wording: 'Thus, some optometrists are not activity involved in making direct referrals using electronic referral platforms or informed of referral outcomes.	Amended as suggested.
Page 3, line 23. A lot has changed since C Roberts blog 2021, consider new reference page https://transform.england.nhs.uk/key-tools-and-info/digital-playbooks/eye-care-digitalplaybook/improve-referrals-to-eye-care-services/	Thank you, the citation has been updated.
Page 3, line 32. The systematic review conclusions demonstrated that optometrist-facilitated teleophthalmology can dramatically reduce referrals and streamline care. In addition, the increasing prevalence of OCT in optometric practice represents an underutilized resource for HES.	We have now included the point of underutilisation.
Page 5, line 6. Still, they expected feedback to explain the reason for non-referrals. The optometrist has made a referral, the HES has reviewed the OCT scan and decided that there is no need for the patient to be seen face to face. This is not a non-referral. Consider rewording: Still, they expected feedback to explain the reason for not being seen for a face-to-face visit.	Reworded as suggested.
Page 5, line 9-45 Comment: some really important messages around timely feedback to patient and referral learning feedback loop to optometrists.	Thank you
Page 6, line 25. Suggest ... reduce unnecessary HES visits and	Amended
Page 6, line 35. Please reference NHS planning guidance 2023/24, where the key actions are to expand direct access and self-referral where GP involvement is not clinically necessary. Integrated care system commissioners are asked to put in place direct referral pathways from community optometrists for all urgent and elective care consultations by September 2023. This is to release capacity in GP practices. NHS England. Priorities and operational-planning-guidance 2023/2024.	We have now included this reference.

https://www.england.nhs.uk/wp-content/uploads/2022/12/PRN00021-23-24-priorities-andoperational-planning-guidance-v1.1.pdf	
Page 7, line 25. Comment: further work should also explore connectivity and interoperability with primary and secondary care systems.	We have now included this.
Reviewer 2	
This is helpful qualitative research as part of a broader trial.	Thank you.
I think the reader needs to be clear from the beginning about the narrow scope of the teleophthalmology in question; ie. name the process as asynchronous triage based on macular multi-modal imaging. Since there are many modes of teleophthalmology, it's helpful for the reader to be more informed of the system being investigated for the context of the patient and clinician experience to follow.	We have reiterated the scope in the introduction and been more specific about the system used in our study.
Small point is long sentences; Pg 3 paragraph beginning line 15 and pg 6 paragraph beginning line 28 as 2 examples leading to loss of clear meaning being conveyed.	Many thanks, these have been amended now.